# A Novel Wheel-Legged Hexapod Robot

**DOI:** 10.3390/biomimetics7040146

**Published:** 2022-09-29

**Authors:** Yong Ni, Li Li, Jiahui Qiu, Yi Sun, Guodong Qin, Qingfei Han, Aihong Ji

**Affiliations:** Lab of Locomotion Bioinspiration and Intelligent Robots, College of Mechanical and Electrical Engineering, Nanjing University of Aeronautics and Astronautics, 29 Yudao Street, Nanjing 210016, China

**Keywords:** wheel-legged robot, motor control, motion gait, biomimetics, trajectory planning

## Abstract

Traditional mobile robots are mainly divided into wheeled robots and legged robots. They have good performance at fast-moving speeds and crossing obstacles, and weak terrain adaptability and moving speeds, respectively. Combining the advantages of these two types mentioned, a multi-functional wheel-legged hexapod robot with strong climbing capacity was designed in this paper. Each wheel-leg of the robot is driven directly by a single motor and can move smoothly and quickly in a diagonal tripod gait. Based on the obstacle-crossing way of the wheel-leg and combined with the characteristics of insects moving stably in nature, the middle part of the robot body is wider than head and tail. Tripod gait was selected to control the robot locomotion. A series of simulations and experiments were conducted to validate its excellent adaptability to various environmental conditions. The robot can traverse rugged, broken, and obstacle-ridden ground and cross rugged surfaces full of obstacles without any terrain sensing or actively controlled adaptation. It can negotiate obstacles of approximately its own height, which is much higher than its centre of gravity range.

## 1. Introduction

Wheel-legged robots are distinct from traditional wheeled or legged robots in that they have arc-shaped legs, which gives them stronger terrain adaptability. Each leg is driven by a separate motor, so their coordinated movement can maintain body steadiness and adapt to complex terrain. Their arc shape ensures that the robot can move at high speeds on soft and rugged terrain.

Researchers in Europe and the United States have developed a series of biomimetic robots with locomotion that resembles insect movement [1]. Some representatives include Robot III and Robot IV developed by Case Western Reserve University, and the RHex bionic robot developed by the University of Michigan, University of California Berkeley and McGill University of Canada [2,3]. RHex is a wheel legged hexapod robot with independent power, non-binding and flexibility. RHex has only six actuators, and each hip joint has a motor. The stability and high mobility of the robot come from a very simple clock driven, open-loop triangular gait. In Asia, the Beijing Institute of Technology has designed a wheel-legged robot [4]. Sun et al. proposed a new type of deformable wheel leg mobile robot, which can be applied to flat and rugged terrain [5]. Peng et al. designed a coordinated control framework to control the wheel leg robot, and adopted impedance control based on the force feedback method to avoid wheel shaking during driving [6]. Cui et al. studied the adaptive optimal control problem of the wheeled legged robot in the absence of an accurate dynamic model. One of the key points is to use reinforcement learning and adaptive dynamic programming to derive a learning based adaptive optimal control solution [7]. Zhao et al. designed and developed an all terrain wheeled retreat hybrid robot with strong adaptability to the environment in order to further improve the ability of wheeled retreat robot to cross obstacles [8]. These bionic robots can perform certain tasks similar to those achieved by insects; however, they have disadvantages such as a tedious control system design, unstable control performance, poor environmental adaptability and low endurance. Additionally, they can only move for a transitory time in the experimental environment. 

In nature, many hexapods, such as crickets and ants, can travel quickly over the ground. This is mainly dependent on a diagonal tripod gait and unique leg attachment structure [9]. The present study proposes a six-wheel-legged robot that can easily navigate diverse terrain similar to that found in natural environments. The robot can perform fast walking in the diagonal tripod gait without demanding perfect battery power, and its energy consumption is relatively low. For instance, it can run at a maximum speed for 60 min on a relatively flat road surface. 

This robot’s design consists of body and wheel-leg components. Each wheel-leg has only one rotational degree of freedom and is actuated by a DC servo motor. The wheel-legs are attached to an output shaft by a 3D-printed self-made coupling. Wheel-legs are used mainly so that this kind of robot can precisely control the ground reaction forces (GRFs) of the legs. Wheels rely on the horizontal component of the GRF when the robot moves forward through friction, while the vertical component has little effect on the motion [10]. The wheel-leg’s foot simultaneously controls the contact angle with the ground, the GRFs and environment adaptability. In this case, it can effectively prevent the slip phenomenon [11]. The straight-rod leg type is more likely to get stuck in mud under complex geological conditions, such as stone seams and soft surfaces. However, wheel-legs can avoid such risks.

In terms of control strategy, a central pattern generator (CPG) was used, which is a neural circuit able to produce periodic outputs without requiring any periodic inputs [12]. It has been widely shown that CPGs can achieve effective animal locomotory gaits [13]. The wheel-legs are driven with reference to the oscillating trajectory detailed in Section 3. The trajectory implements alternating triangular gaits in a closed-loop control manner. Each movement cycle is divided into two phases: fast and slow. The slow phase is used when the robot is assumed to be in contact with the ground. When the legs are off the ground, the robot enters the fast phase, which enables the group of legs to touch the ground in time before other wheel-legs leave the ground. In this way, unsupported phase falls can be prevented [14,15,16].

The present study describes several attempts to design wheel-legged crawling robots. The selection of the robot’s mechanical structure and control strategy is described in detail. On this basis, the robot can use a variety of locomotory modes, such as walking, running, obstacle avoidance and climbing stairs. This provides a good research platform for future ground-crawling robots.

## 2. Structure and Model

### 2.1. Mechanical Design

In the process of robot design, assembly failures can be frequent due to the complicated mechanisms required. Hence, we used 3D additive manufacturing technology to simplify the mechanical structure and improve its robustness [17]. The overall design is shown in Figure 1. The robot consists of a body and six wheel-legs, which are independently controlled by six motors. Each wheel-leg has only one rotational degree of freedom. The motor is fixed on the body by a motor seat and has an output shaft with a self-made coupling for fixing the wheel legs. The bottom plate of the fuselage is made of aluminium sheet. This plate, coupling and wheel legs are all fabricated by 3D printing. This simple manufacturing process ensures that a set of finished products can be quickly reproduced. Besides, convenient assembly and disassembly is beneficial to further improve the device. Different from RHex robot, we have installed a number of adhesive foot pads on the legs of the robot, which is conducive to strengthening the friction between the robot and the surface during movement, reducing the relative sliding, and further enhancing the adaptability of the robot to the moving environment.

Six curved wheel-legs are symmetrically mounted on both sides of the robot body. The front and rear wheel-legs are contracted by a certain distance (slightly larger than the leg width). In this way, the width of the front and rear of the body can be reduced to decrease the ratio of the mass of the body to that of the whole machine. It also solves the interference problem that occurs when all three legs on one side are moving. This design reduces the length of the body and makes the movement of the robot gait easier to implement. In this configuration, the robot travels in a diagonal tripod gait to ensure stable movement. Compared to a four-footed robot, the hexapod robot possesses a more redundant supporting phase at the station timing, which also provides potential for exploring jumping and climbing movements in the future. It also prevents the robot from falling over due to a too-short support phase. Moreover, since the wheel-legs are curved, they will not collide when moving in opposite directions, so there is no restriction on the direction of advancement or the use of differential-speed steering.

### 2.2. Hexapod Model

In order to calculate and analyse the structure of the robot design, we established the dynamic model shown in Figure 2. This model can also be used to verify the rationality and feasibility of the structure through calculation [18]. The initial state of the robot is set such that the body lies flat on the ground (contact surface) with the six wheel-legs also in contact with the ground. When the power is turned on, the six wheel-legs first slowly rotate so that the body stands up at a relatively slow speed. At this time, the motor output torque is mainly used to overcome the work of gravity. The maximum motor drive torque ($T_{\max}$) required for each leg is
(1)$$T_{\max}=\frac{G}{6}\times \sqrt{r_{\max}{}^{2}-h^{2}}$$

According to this model, the distance r between the contact point and rotating axis varies due to the curvature of the wheel-legs. For calculation, we analyse the maximum value to account for all possible cases.

f is the leg’s frictional force on the contact point; θ is the angle between the frictional force f and the negative direction of the *Y*-axis; and *F* is the normal force of the *Z*-axis. These parameters can be used to calculate the motor output torque.

The proposed robot imitates an insect’s diagonal tripod gait for advancement. The six legs are divided into two groups that are symmetrically distributed in a triangle during the travelling process. One group performs the support phase while the other is in the swing phase. They alternate phases to ensure that at least one set of legs can always make contact with the ground to prevent the body from falling. Changing the length of contact between the legs and the ground can control the forward speed of the robot. For turning, the differential-speed steering mode is used to adjust the support and vacating time of the in-phase legs. The mathematical model in Equation (2) shows that we introduce a frictional force in the non-forward direction. We specify an angle *θ* on the model, which is the angle between the frictional force f and the negative direction of the *Y*-axis. When the robot moves forward, the torque ($T_{i}$) required by the motor is
(2)$$T_{i}=f_{i}\cdot \cos\theta_{i}\cdot r_{i}$$

The theoretical values of the motor parameters can be obtained via the above analysis, include the rotational speed of the motors, as shown in Table 1. 

## 3. Control Strategy

The control block diagram of the robot is shown in Figure 3. The main chip in the block is an STM32F407, which is used to control the drive circuit by generating a PWM wave. An optocoupler is used for stimulating signal isolation. The main control board is separated from the motor drive circuit to protect the main control chip. The motor drive circuit contains six independent H-bridge high-voltage, high-current, dual-bridge drivers that can receive standard TTL logic-level signals and drive six motors of 24 V or less. The motor model is a MAXON RE25 DC servo motor. The planetary reduction gearbox has a reduction ratio of 46:1, and its continuous output torque can reach 12 Nm, which meets the needs of the robot perfectly. An incremental encoder is mounted on the motor to feed back the output pulse signal. The host computer software communicates with the main chip through a serial port and can obtain the real-time pulse number. After that, it can convert the pulse number into a corresponding angle value to monitor the wheel-legs’ motion in real-time. Once a problem arises, it can be instantly corrected. A power module transforms the battery voltage as needed for the different components of the robot.

The robot’s wheel-legs adopt a diagonal tripod gait. To increase motion stability, the rotation cycle of the wheel-legs is divided into two phases: fast and slow. These two phases alternate with each other to ensure that at least one set of wheel-legs touches the ground at any time [19,20]. This avoids collision between the robot body and the ground. Each leg performs a periodic desired trajectory while travelling and is controlled by a proportion integration (PI) control. The control scheme is shown in Figure 4, which shows that the rotation position control of the wheel-leg output shaft consists of three closed-loop controls which, from inside to outside, are the current loop, speed loop and position loop.

The rotation speed of the robot wheel leg is realized by changing the duty cycle of PWM wave. PWM is the abbreviation of pulse width modulation, that is, by modulating the width of a series of pulses, the required waveform (including shape and amplitude) can be obtained equivalently. We set the signal output cycle as T, and control the output power by adjusting the proportion of working time in each cycle to the total cycle (duty cycle D).

We adopt proportional integration (PI) parameter adjustment control. In PI control, the integral term (I) in the controller output is proportional to the current error value and the accumulated value of past error values, so the integral effect itself has a certain lag, which is unfavorable to the stability of the system. If the coefficient of the integral term is not set well, its negative effect can hardly be quickly corrected by the integral action itself. The proportional term (P) has no delay. As long as the error occurs, the proportional part will take effect immediately. The PI controller not only overcomes the shortcomings of simple proportional regulation with steady-state error, but also avoids the shortcomings of simple integral regulation with slow response and poor dynamic performance. As shown in Figure 4, the input of the current loop is the output of the speed loop after PI regulation, and the difference between the input value of the current loop and the feedback value of the current loop is PI regulated and output to the motor in the current loop. The input of the speed loop is the output after PI regulation of the position loop and the feedforward value of the position setting. The difference between the input value of the speed loop and the feedback value of the speed loop is output to the current loop after PI regulation of the speed loop. The feedback of the speed loop comes from the feedback value of the encoder obtained through the speed solver. Speed loop control includes speed loop and current loop. The input of the position ring is the external pulse, which is used as the “setting of the position ring” after smooth filtering and electronic gear calculation. The input value of the position loop and the pulse signal fed back from the encoder are calculated by the deviation counter. After PI adjustment of the position loop, the sum of the output and the feedforward value given by the position constitutes the given value of the speed loop. The feedback of the position loop also comes from the encoder.

### 3.1. Forward Alternating Tripod Gait

A schematic diagram of the trajectory of a wheel-leg from its initial point is shown in Figure 5A. Since we use an incremental encoder, we can simply determine the relative rotational angle of the wheel-leg by conversion [21,22]. Therefore, the 0 position means nothing or the legs’ positions when the robot lies flat on the ground. As the tripod gait control trajectory of each leg is a periodic function with respect to time, we define $X=[\Phi_{0},T,t_{s},\Phi_{s}]$ as the motion control parameter. In a single cycle, both tripods go through slow and fast swing phases, covering $\varphi_{s}$ and $2\pi -\varphi_{s}$ of a complete rotation, respectively. $\Phi_{0}$ is the required rotation angle of the robot from lying to stand, T is the period in which a single leg finishes one turn, $t_{s}$ is the duration of the slow phase, and $\Phi_{s}$ is the angle through which the slow phase turns.

Figure 5B illustrates the prescribed trajectory in a more intuitive and understandable way. In this figure, $t_{t}$ is the time at which all six legs are supported at the same time. To prevent the centre of gravity from going up and down, we maintain the condition $t_{t}\geq 0$. During movement of the robot, the parameters of the above X are modified to achieve the corresponding desired trajectory, thus obtaining the corresponding motion control.

### 3.2. Turning

In the steering control strategy, the common differential-speed steering method is used along the desired trajectory with the same progressive state. The main advantage of differential-speed steering is only the speed of the tripod gaits needs to vary. That is, the left and right tripod gaits adopt different speeds, even reversing to achieve steerage. The same group of three feet is still guaranteed to have internal synchronization. Similar to forward motion control, we ensure that at least three legs are on the ground at any time. The rotational speed depends on the adjustment of parameter *X*.

In contrast to the forward motion parameter, we add disturbances to the forward motion control parameters of the contralateral leg to achieve steering during forward motion. T and $\Phi_{s}$ remain constant, so we only modify the time of $t_{s}$, add a time parameter $\Delta t_{s}$, and use $Xl=[\Phi_{0},T,t_{s}+\Delta t_{s},\Phi_{s}]$ and $Xr=[\Phi_{0},T,t_{s}+\Delta t_{s},\Phi_{s}]$ to denote the wheel legs on the left and right sides, respectively, when turning.

### 3.3. Jumping

When moving in tripod gait, the wheel-legged robot can achieve crawling and obstacle-blocking on any rough surface. However, if the robot encounters a damaged road section, such as a gully or a small crack in the course of performing a task, the tripod gait will not work. To enable the robot to leap over such obstacles, we designed a jump gait that mimics the jumping mechanism of locusts. As shown in Figure 6, the front leg remains unchanged during the entire jumping phase. The jump is split into two phases: (1) the middle legs start rotating while the hind legs remain stationary, raising the front of the robot to a 45° angle and; and (2) the hind legs simultaneously rotate at a relatively fast angular velocity, accelerating the robot in the direction of the body.

## 4. Simulation and Experimental Studies

### 4.1. Simulation Studies

In this section, simulations are described that used the kinetic and actuator models described previously to demonstrate the feasibility of the basic motion of our design within practical driving limitations. To measure the motion control parameter X (Section 3) more accurately and observe the relationship between the legs of the robot during the diagonal tripod gait, we used V-REP to simulate and analyse the robot to determine the reasonable control parameters, as shown in Figure 7. The simulation shows that the maximum amplitude of the centre of gravity is only 25 mm when the robot is moving over the ground without any obstacles. This amplitude is much smaller than its own height, which proves that this robot can run smoothly.

### 4.2. Experimental Verification

The experimental six-wheel-legged robot platform, which combines the structural and control system designs, is shown in Figure 8. The body size of the robot is 50 cm (length) × 40 cm (width) × 13.5 cm (height of the axle in an upright state), and its weight is 4.34 kg. 

We placed robots on different experimental surfaces, such as wooden floors, marble pavements, water-filled marble pavements, flat grass, high grass, and rough surfaces to carry out several experiments. All the experiments used the same gait parameters of X=[120,2.175,90,1.5]. Each group was set to crawl at least 10 times to record the success rate and analyse the forms of failure, which include hardware circuit problems, deviations from the expected track, operational errors and stuck wheel-legs. The experimental situation is shown in Table 2. Taking crawling data from 10 successful runs on various surfaces, the average speed and speed interval of each surface were obtained. As shown in Figure 9, the robot moved steadily on the indoor and outdoor surfaces with speeds varying from 0.189 m/s to 0.216 m/s.

As shown in the simulation, turning motion is achievable. Although the robot has no extra degrees of freedom, circular motion by differential steering can be achieved by varying the speeds of the different wheel-legs. Using a right turn as an example, Figure 10 shows the action sequence of the robot during the turning experiment and Figure 11 shows the deflection direction of the robot’s turning gait; that is, the speed curve in the direction to the right of the forward direction. In the initial stage, the offset is not obvious, but after two gait cycles, as the deflection angle increases, the speed increases continuously and a 90° turn can be achieved after about 10 s of motion.

We also tested the robot’s obstacle height limit. We conducted the experiment of the limit obstacle height on the stairs as shown in Figure 12. Take the first step as the obstacle crossing target, and the height of the step foundation is 11 cm. Add wood on it to set the obstacle crossing height. The thickness of each board is 1.7 cm. If it can be climbed, one board will be added. Three experiments shall be conducted for each height. If the robot can climb over two or more times, it shall be deemed as successful. Otherwise, it shall be deemed as failure. After failure, the object with a thickness of less than 1.7 cm shall be placed between the boards to reduce the limit height of obstacle climbing until the final height is measured. The experimental results are shown in Table 3. The experiments show that the maximum height of obstacle a robot can climb is approximately 22.2 cm. The ratio of this height limit to the robot’s upright height (13.5 cm) is about 1.63. The main cause of failure is slipping off obstacles, which occurs when the robot fails to hook onto an obstacle. Hence, the non-slip material attached to the wheel-legs needs to be improved in the future. Another cause of failure is that the robot’s wheel-legs and obstacle contact points are too close to the shaft, which will cause the motor output torque to struggle to meet the demand. This suggests another improvement for the robot: adding a vision system to the front. Then, the robot could reasonably plan its gait according to the position and height of an approaching obstacle so as to ensure that it touches the obstacle at the roots of its legs as much as possible. In this way, the advantage of the high torque of the motor is fully utilized when the obstacle is overcome.

Before studying the jumping gait, we tested the longest gully length that the robot can cross under a normal gait. The experimental method is shown in Figure 13. The results demonstrate that when the robot encounters a surface with a gap while advancing and cannot cross the gap in the normal forward state, it can jump over it. The experimental results in Table 4 show that the maximum gap distance that the robot can cross is 18 cm. To ensure a safety margin, the robot needs to adopt a jumping gait when the maximum road damage distance exceeds 16 cm.

To verify the jumping performance of the robot, we use the jumping principle discussed in Section 3.3. First, we fix the two front legs and then drive the middle two legs to rotate 120°. After raising the body, the middle and rear legs are adjusted to full speed. The jumping experiment process is shown in Figure 14. Then, the robot will rotate to obtain jump acceleration. The jump distance of the robot in this mode is 44 cm, which is 0.88-times the length of its body.

The experiments discussed above demonstrate that the wheel-legged robot can perfectly achieve the behaviours of standing up, crawling, turning, overcoming obstacles and jumping. It meets the design requirements and performs well in both flexibility and moving speed. From the results of many experiments, the maximum height of obstacle that the robot can climb is 22 cm, which is even higher than itself. By adding multiple sensors and a digital image processing system, the wheel-legged robot could achieve the functions of obstacle avoidance and special reconnaissance in a stable and reliable way.

## 5. Conclusions

This paper describes the design and manufacture of a new type of wheel-legged crawling robot. Its small size and light weight allow it to overcome obstacles on complex surfaces. The maximum height of obstacle that the robot can climb is much higher than the maximum amplitude of the centre of gravity. Under certain loading, it can still realize autonomous obstacle avoidance using a diagonal triangle gait. Nevertheless, its flexibility and robustness are much poorer than those of living animals. We believe that further systematic application of certain animal operating principles will help achieve significant improvements in performance and provide more information for the development of robot designs.

## Figures and Tables

**Figure 1 biomimetics-07-00146-f001:**
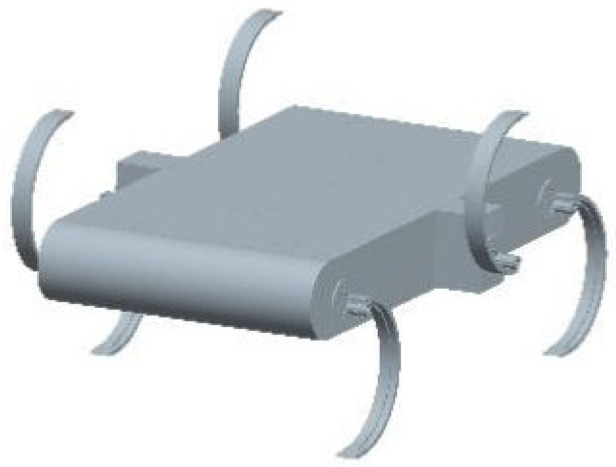
Overall design of the robot.

**Figure 2 biomimetics-07-00146-f002:**
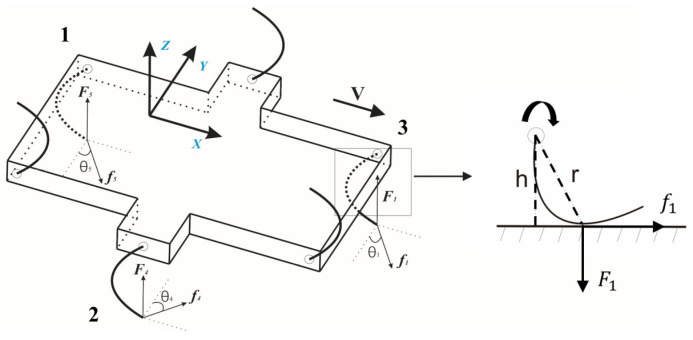
Mathematical stress analysis of the robot.

**Figure 3 biomimetics-07-00146-f003:**
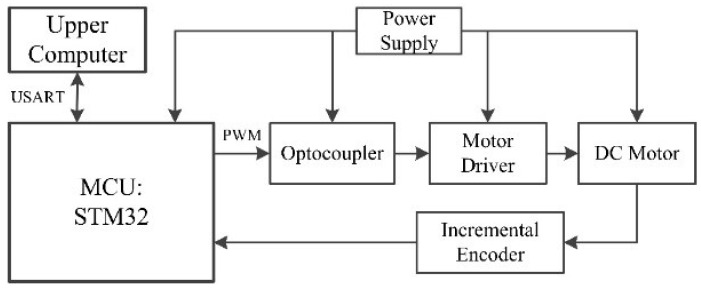
Control block diagram of the robot.

**Figure 4 biomimetics-07-00146-f004:**
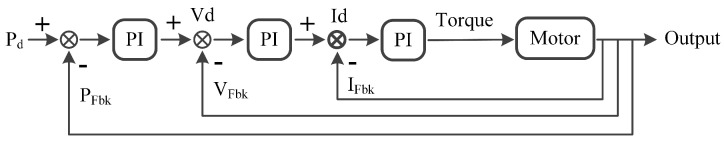
Schematic of the PI parameter adjustment process.

**Figure 5 biomimetics-07-00146-f005:**
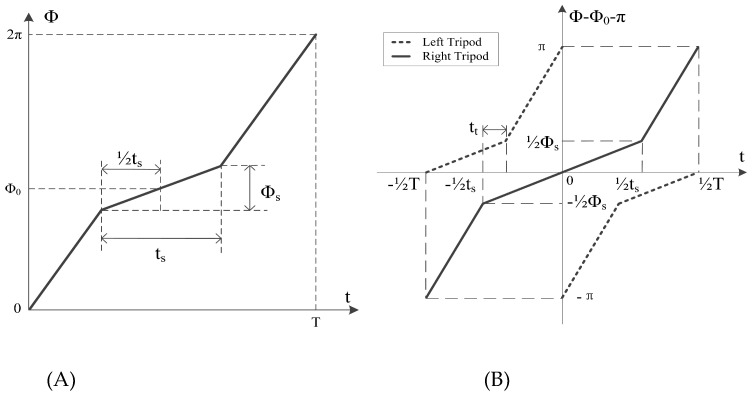
Wheel-leg trajectory planning when moving forward. (**A**) Schematic diagram of the trajectory of a wheel-leg from the initial point; (**B**) expected curve of trajectory planning, where T is the period of a rotation of each foot, and ϕ is the rotation angle at the corresponding moment. Gait planning is carried out for each foot of the hexapod robot according to the given gait period. The robot can obtain a stable and reliable running gait.

**Figure 6 biomimetics-07-00146-f006:**
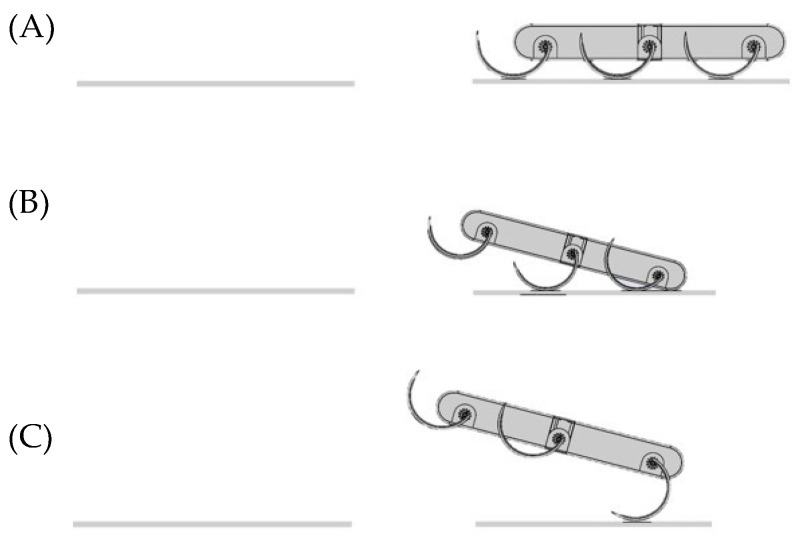
Mechanism of the robot’s jumping gait. In the first stage (**A**), the robot’s centre of gravity is lowered before jumping. In phase two (**B**), the leg rotates to raise the back of the robot to an angle of 45°. In stage three (**C**), the middle and rear legs rotate simultaneously at a relatively fast angular speed, giving the robot a slanting acceleration along the direction of the body.

**Figure 7 biomimetics-07-00146-f007:**
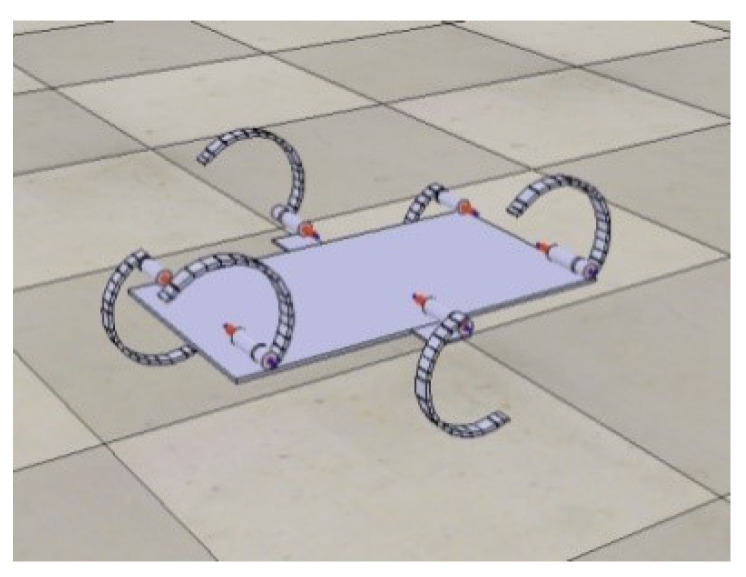
V-REP simulation of the robot.

**Figure 8 biomimetics-07-00146-f008:**
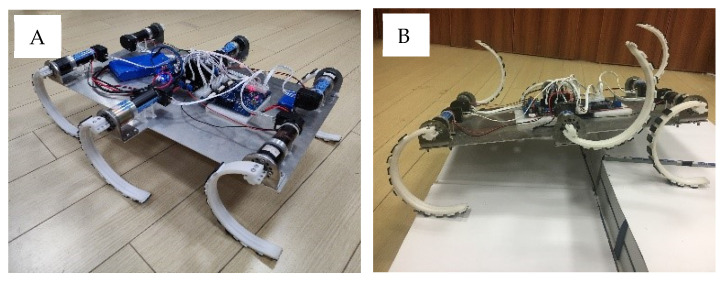
Wheel-legged robot experimental platform.

**Figure 9 biomimetics-07-00146-f009:**
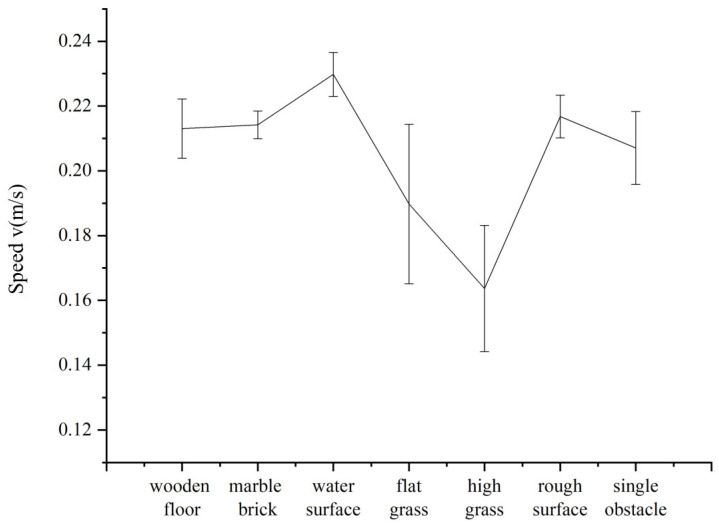
Comparison of average forward speeds on different surfaces.

**Figure 10 biomimetics-07-00146-f010:**
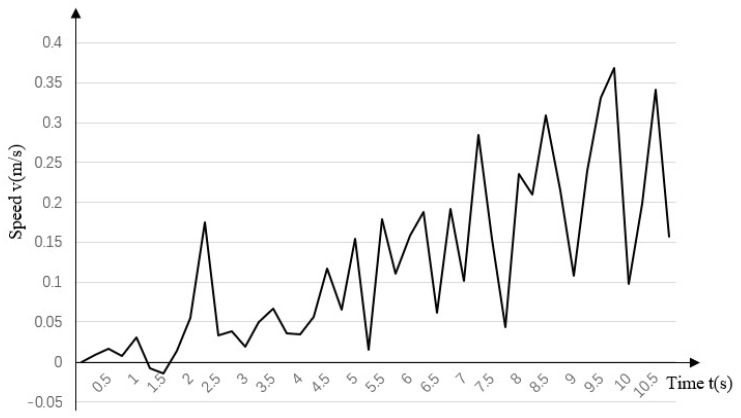
Yaw speed curve during turning gait.

**Figure 11 biomimetics-07-00146-f011:**
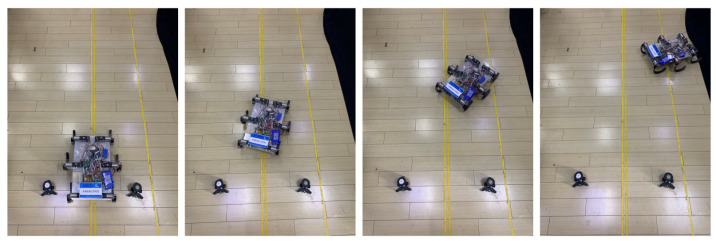
Turning experiment process of robot.

**Figure 12 biomimetics-07-00146-f012:**
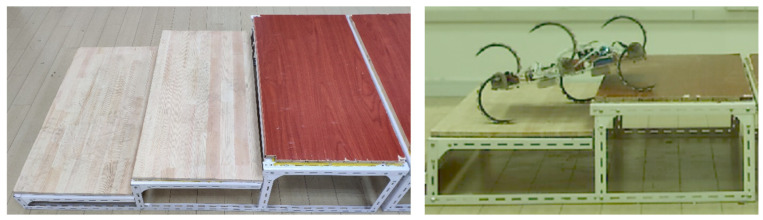
Experiment of robot crossing gullies.

**Figure 13 biomimetics-07-00146-f013:**
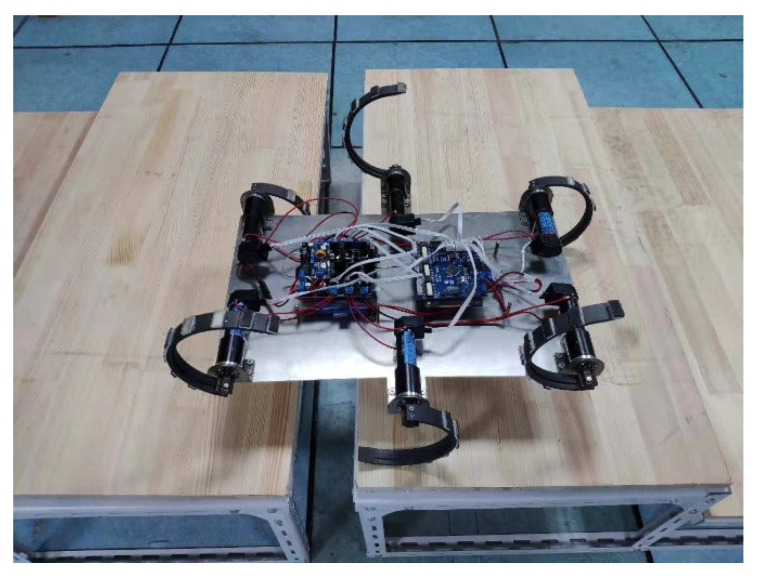
Experiment of robot crossing gullies.

**Figure 14 biomimetics-07-00146-f014:**
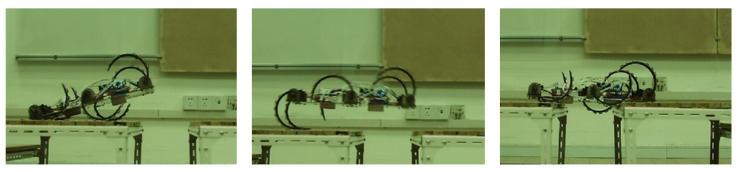
The jumping experiment process of the robot.

**Table 1 biomimetics-07-00146-t001:** List of theoretical motor parameters.

Gait	Parameter	1	2	3
Starting gait	Torque (mN·m)	69	69	69
	Speed (r·min^−1^)	16.7	16.7	16.7
Forward alternating tripod gait	Torque (mN·m)	41	276	164
	Speed (r·min^−1^)	10	66.7	40

**Table 2 biomimetics-07-00146-t002:** Robot straight-line test results on different surfaces.

Parameter	Wooden Floor	Marble Brick	Water Surface	Flat Grass	High Grass	Rough Surface	Single Obstacle	Multi-Obstacle
Number of runs	10	10	10	12	13	11	12	17
Successful runs	10	10	10	10	10	10	10	10
Success rate	100%	100%	100%	83.3%	76.9%	90.9%	83.3%	58.9%
Hardware circuit	/	/	/	/	/	/	/	1
Deviation from expected orbit	/	/	/	2	1	/	1	3
Operating errors	/	/	/	/	2	1	1	1
Stuck wheel-legs	/	/	/	/	/	/	/	2

**Table 3 biomimetics-07-00146-t003:** Experimental results of robot obstacle-climbing task.

Obstacle Height (cm)	Successful Runs	Reason for Failure
12.7	3	/
14.4	3	/
16.3	3	/
18.4	3	/
20	2	Low torque
21.5	3	/
21.7	2	Not hooked
22.2	1	Not hooked
22.8	0	Not hooked

**Table 4 biomimetics-07-00146-t004:** Experimental results of robot jumping task.

Gully Width (cm)	Successful Runs	Reason for Failure
6	3	/
8	3	/
10	3	/
12	3	/
13	3	/
15	3	/
16	3	/
17	2	Wheel-leg stuck
18	1	Hind legs failed to touch ground, body leaned back
19	0	Distance too far

## Data Availability

Not applicable.

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
