# Peer review of "A Novel Wheel-Legged Hexapod Robot"

_biomimetics, 2022, doi:10.3390/biomimetics7040146_

Round 1
Reviewer 1 Report
A multi-functional hexapod robot with climbing capacity has been brought out in this paper. To be a better paper, the following problems should be revised:
1. The abstract of this paper only introduce a six-wheel-legged hexapod robot, but which should introduce the research methods and routes briefly, so I think the abstract should be rewritten.
2. The article structure should be adjusted, such as “Mechanical Design” and “Hexapod Model” should not be two parts of “Introduction”.
3. The mechanical design of the robot is similar with a robot of Boston Dynamics “rhex”, so what is the novelty of this paper?
4. Page 6, line 4, “The mathematical model in Equation 2 shows that we introduce a frictional force in the non-forward direction.” Equation 2 or Figure 2? Or is there another equation not indicated?
5. In FIGURE 9, label the y-axis(speed, units). It should be a column graph as it shows categories.
6. FIGURE 10 reflects the change of right turn speed, but it cannot visually show the behavior of right turn. Maybe it's better to show it with action sequence diagram.
Author Response
Thanks to the editorial department for the guidance of the manuscript. Please see the attachment.

Reviewer 2 Report
This paper proposes a wheel-legged hexapod robot with strong climbing capacity. Combining the advantages of legged robots and wheeled robots, the robot can traverse various terrains. It‘s a good work.
The following comments may be helpful in improving the manuscript:
1. Some most recent developments in the field of wheel-legged robot should be added in Introduction, and the references should be updated.
2. In section 1.2, there is a minor error in equation (1), authors should check all the equations carefully. In addition, what’s the relationship between equation (1) and equation (2)? And what’s the direction of the friction force?
3. How the theoretical values in Table 1 were obtained need to be described clear. And the parameter ‘1, 2 and 3’ in Table should be marked out in Figure 2.
4. As described in line 172, incremental encoder was used to detect the rotational angle of the wheel-leg, is the absolute 0 position of the wheel-leg necessary in the control strategy? If so, how to determine the absolute 0 position of the wheel-leg?
5. Many climbing and jumping experiments of the robot have been conducted. It could be better if there are more details of the experiments.
Author Response
Thanks to the editorial department for the guidance of the manuscript.Please see the attachment.

Round 2
Reviewer 1 Report
The paper has been revised carefully and be better than v1. But there are some still problems should be resolved.
1. The promotion of the paper robot compared with RHex should be described in this paper, because it’s similar with the paper robot.
2. The focus of this paper is to design the motion control strategy of the robot in the face of a variety of different complex moving surfaces, but there are few details about the control method for the robot’s adaptability to terrains, which should be reinforced.
Reviewer 2 Report
The authors have made modifications according to the comments. However, there are some problems to be further improved as well:
(1) The authors have added some recent references. All the reference numbers cited in the manuscript should be checked and updated.
(2) In Table 1, the word ‘starting gait’ was repeated, and what’s the meaning of ‘starting gait’.
(3) Line 226, ‘Section 2’ should be ‘Section 3’. And the article should be checked thoroughly.
